# Complete Genome Sequence of Herpes Simplex Virus 2 Strain G

**DOI:** 10.3390/v14030536

**Published:** 2022-03-05

**Authors:** Weizhong Chang, Xiaoli Jiao, Hongyan Sui, Suranjana Goswami, Brad T. Sherman, Caroline Fromont, Juan Manuel Caravaca, Bao Tran, Tomozumi Imamichi

**Affiliations:** 1Laboratory of Human Retrovirology and lmmunoinformatics, Frederick National Laboratory for Cancer Research, Frederick, MD 21702, USA; jiaoxl@protonmail.com (X.J.); hongyan.sui@nih.gov (H.S.); suranjana.goswami@nih.gov (S.G.); bsherman@mail.nih.gov (B.T.S.); 2Sequencing Facility, Cancer Research Technology Program, Frederick National Laboratory for Cancer Research, Frederick, MD 21702, USA; caroline.fromont@nih.gov (C.F.); juanmanuel.carava@nih.gov (J.M.C.); tranb2@mail.nih.gov (B.T.)

**Keywords:** HSV-2, complete genome sequence, ‘α’ sequence, packaging signals, genome termini, strain G

## Abstract

Herpes simplex virus type *2* (HSV-2) is a common causative agent of genital tract infections. Moreover, HSV-2 and HIV infection can mutually increase the risk of acquiring another virus infection. Due to the high GC content and highly repetitive regions in HSV-2 genomes, only the genomes of four strains have been completely sequenced (HG52, 333, SD90e, and MS). Strain G is commonly used for HSV-2 research, but only a partial genome sequence has been assembled with Illumina sequencing reads. In the current study, we de novo assembled and annotated the complete genome of strain G using PacBio long sequencing reads, which can span the repetitive regions, analyzed the ‘α’ sequence, which plays key roles in HSV-2 genome circulation, replication, cleavage, and packaging of progeny viral DNA, identified the packaging signals homologous to HSV-1 within the ‘α’ sequence, and determined both termini of the linear genome and cleavage site for the process of concatemeric HSV-2 DNA produced via rolling-circle replication. In addition, using Oxford Nanopore Technology sequencing reads, we visualized four HSV-2 genome isomers at the nucleotide level for the first time. Furthermore, the coding sequences of HSV-2 strain G have been compared with those of HG52, 333, and MS. Moreover, phylogenetic analysis of strain G and other diverse HSV-2 strains has been conducted to determine their evolutionary relationship. The results will aid clinical research and treatment development of HSV-2.

## 1. Introduction

Herpes simplex virus type *2* (HSV-2) is a member of the *Alphaherpesvirinae* subfamily of the *Herpesviridae* family [1]. It is commonly found as the causative agent of genital tract infections. It has been reported that 11.9% of persons aged 14 to 49 years (12.1% when adjusted for age) in the US [2] and 13.2% of the world’s population aged 15–49 years [3] have HSV-2 infection. The risk of HIV acquisition has been found to be increased in individuals with HSV-2 infection [4,5,6]. It has also been found that coinfection of HSV-2 with HIV can increase the shedding of HIV particles and result in a higher risk of HIV transmission to sex partners [7]. Another challenge of HSV infection treatment and control is the periodic reactivation of latent virus [8,9,10]. HSV-2 strain G (HSV-2 G) was originally isolated from a patient in Chicago in the 1960s [11]. It has been commonly used for HSV-2 research [12,13,14] and vaccine development [15,16]. Complete sequencing of the HSV-2 G genome will further allow us to understand HSV-2 structure and virological function and help us control and treat the infection of this virus. 

HSV has a linear double-stranded DNA genome that contains GC rich regions and multiple repeat elements, including two copies of long-inverted repeat-R_L_ (TR_L_ and IR_L_), two copies of short-inverted repeat-R_S_ (IR_S_ and TR_S_), and three ‘α’ sequence regions, two at both ends of the genome in direct orientation and one at the internal L-S junction in an inverted orientation. Two unique regions (U_L_ and U_S_) are bracketed by inverted repeats: U_L_ is bracketed by TR_L_ and IR_L_; U_S_ is bracketed by IR_S_ and TR_S_ [17]. Soon after infection, the linear HSV genome circularizes [18,19], DNA replication is initiated at the origin with theta mechanism, then switches to rolling circle mode, leading to the production of concatemeric HSV DNA [20]. The concatemeric HSV DNA can be further replicated from the origin and form the branched structures [21]. The ‘α’ sequence plays roles in HSV genome DNA circulation and replication [17,18] and cleavage of the concatemeric HSV DNA and packaging of the processed HSV genome DNA [22]. The genome isomerization adds additional complexity to the HSV genome leading to four isomeric arrangements of viral DNA [23,24,25], and the ‘α’ sequence was demonstrated being involved in this process [26]. 

High GC content, highly repetitive regions, and complicated genome structure pose challenges to sequencing the HSV whole genome with traditional Sanger sequencing technology and next-generation sequencing platforms since the sequencing reads from these platforms cannot span the repetitive regions. As of 8 December 2021, a total of 115 HSV-2 genome sequences could be found in the NCBI Nucleotide database, including complete or partial sequences. Among them, only a limited number are complete or near-complete genome sequences: HG52 [27], 333 [28], and MS [29]. The low passage strain SD90e genome has been sequenced with Illumina and 454 platforms, and assembled with HG52 as the reference to fill the gap [30]. Only a partial genome sequence of HSV-2 G strain (NCBI Accession: KU310668.1) was available, which was sequenced with the Illumina platform [16]. Long read DNA sequencing technologies, including SMRT sequencing from Pacific Bioscience (PacBio) and Nanopore sequencing from Oxford Nanopore Technologies (ONT), provide the opportunity to resolve these long repetitive regions in HSV genomes since these sequencing reads can span the repetitive regions. The strain 333 genome sequence mentioned above was sequenced with the PacBio platform, and the strain MS sequence mentioned above was sequenced with a combination of long sequencing reads from the PacBio platform and short sequencing reads from the Illumina platform. Since sequencing errors in PacBio sequencing reads are random, the genome assembly with the PacBio platform can reach high accuracy [31,32]. Our group has successfully sequenced and assembled the full-length genomes of two HSV-1 strains: Maclntyre [33] and McKrae [34]. In this study, we sequenced and assembled the genome of HSV-2 G and compared it with the genome sequences of other strains from different locations. We also determined the termini of the linear genome, analyzed the ‘α’ sequence, and identified the packaging signals and cleavage site within the ‘α’ sequence. Furthermore, with ONT sequencing reads spanning IR_L_ and IR_S_, we have observed four isomers of the HSV-2 genome for the first time at the nucleotide level. 

## 2. Materials and Methods

### 2.1. HSV-2 Strain G Virus and Whole Viral Genome DNA Sequencing

HSV-2 G was obtained from Advanced Biotechnologies Inc (Eldersburg, MD, USA). The virus was propagated in Vero cells (ATCC, Gaithersburg, MD, USA). The genomic DNA was isolated using the PureLink Viral DNA Minikit (Invitrogen, Carlsbad, CA, USA), following the manufacturer’s protocol. A total of 1000 ng of viral genomic DNA was sheared to 10–12 kb using the Megaruptor 3 (Diagenode, Inc., Denville, NJ, USA). A total of 482 ng of sheared DNA was used to prepare a SMRTbell template library with the SMRTbell Express Template Prep Kit 2.0 (Pacific Biosciences, Menlo Park, CA, USA) following the microbial library protocol (Procedure & Checklist—Preparing Multiplexed Microbial Libraries Using SMRTbell^®^ Express Template Prep Kit 2.0). Fragments shorter than 3 kb were removed from the library using a bead size selection method with the standard protocol from the manufacturer (Pacific Bioscience Inc.). Removal of shorter insert fragments that would be preferentially sequenced if present helps achieve a larger subread mean length which can aid assembly as well as improve total sequencing yield. Sequencing primer v4 was annealed, and Sequel II polymerase 2.0 was bound to the library prior to loading on one SMRT Cell 8M on the Sequel II System using diffusion loading. Sequencing was performed with 2 h pre-extension and a 30 h movie. The whole-genome sequencing samples for ONT were prepared using Kit SQK-LSK109 and sequenced on an FLO-MIN106 flowcell of GridION (Oxford Nanopore Technologies, Oxford, UK). Flowcells were run for 65 h to generate the data with default settings, including mux scans every 90 min. The run was basecalled using MinKnow version 21.02.5, which uses Guppy version 4.3.4. At the end of the run, more than 2.5 gb of data were generated per flowcell with a median Phred quality score of about 11.5 with N50 greater than 9 kb.

### 2.2. Whole Viral Genome Assembly

PacBio subreads were assembled into contigs using the PacBio de novo assembler Hierarchical Genome Assembly Process v4.0 (HGAP4) [32] (default settings were used, except that the seed length cutoff was set to 15 kb, and the minimum required alignment length was set to 500 bp). The contigs were aligned against the genome sequence of HSV-2 reference strain HG52 (NCBI Accession: NC_001798.2) [27] by BLASR (v2.0.0) [35] with default settings, and the contigs jointly covering the full length of the HG52 reference genome were merged into a draft sequence and then improved by the PacBio genome resequencing pipeline (SMRT^®^ Analysis v6.0) with default settings (Pacific Bioscience Inc.).

### 2.3. Determination of Genome Termini of HSV-2 Strain G

The PacBio sequencing reads aligning uniquely to the two termini were selected in three steps. (1) The filtered CCS sequencing reads (≥1000 nt) were mapped to the finished HSV-2 G genome sequence with the “Mapping” pipeline in SMRTLink (v.8.0). (2) The IDs of the reads aligning uniquely to the two termini were retrieved from the alignment file with Samtools [36]. (3) These reads were then extracted from the CCS fastq file using SeqKit [37] based on read ID. This subset of reads was then aligned to the finished genome sequence with Minimap2 [38] and visualized with IGV [39]. The finished HSV-2 sequence termini were based on this analysis.

### 2.4. Identification of Sequencing Reads from Four HSV-2 Genome Isomers

IR_L_ and IR_S_ and 1000 bps of their flanking upstream gene UL56 (or UL1) and downstream gene US1 (or US12) were retrieved from genome sequences as the reference sequences for alignment of the ONT sequencing reads with length greater than 17,000 bps using BLASR [35] and to identify those that support the 4 individual isomer types. The reads that cover the full-length of the IR (IR_L_ and IRs) region and at least 500 bps of both the UL and US genes with identity greater than 0.8 were counted as full-length reads supporting the individual isomer type.

### 2.5. Analysis of ‘α’ Sequences

The ‘α’ sequences between IR_L_ and IR_S_ from 10 HSV-2 strains from NCBI GenBank and strain G from this study were retrieved and aligned with ClustalW with standard parameters for DNA sequences. These 10 strains include 2007-38606 (NCBI Accession: KX574873.2), SD90e (NCBI Accession: KF781518.1), HG52 (NCBI Accession: NC_001798.2), CT_sample9e (NCBI Accession: MT364888.1), Sample16 (NCBI Accession: MF564036.1), H1226 (NCBI Accession: KY922720.1), 333 (NCBI Accession: LS480640.1), H12212 (NCBI Accession: KY922726.1), MS (NCBI Accession: MK855052.1), and H12211 (NCBI Accession: KY922725.1). The alignment was then compared with HSV-1 ‘α’ sequences to identify homologous packaging signal motifs [40].

### 2.6. Determination of ‘α‘ Sequence Copy Number in HSV2 Strain G Genome

The PacBio CCS reads aligned to ‘α’ sequence with mapped length >200 bps were identified. To further determine which reads are in TR_L_, TRs, or IR regions, we aligned these 927 reads to coding sequences (CDS) of RL1 and RS1. Reads containing both RL1 and RS1 with mapping length >200 bps are considered as IR reads because it joins the internal repeat region R_L_ and Rs. Reads containing only RL1 (>200 bps) but not RS1 and starting or ending with a nearly full-length ‘α’ sequence (>360 bps) are considered as TR_L_ reads and similarly, reads containing only RS1 (>200 bps) but not RL1 and starting or ending with a TRs ‘α’ sequence (349 bps) are considered as TRs reads.

### 2.7. Sequence Comparison and Phylogenetic Analysis

The CDS of HSV-2 G were aligned with CDS of strain HG52, 333, and MS separately using BLASR, and the insertion, deletion, and substitution variants of each CDS were summarized.

To analyze the relationship of HSV-2 G and other HSV-2 strains, we performed phylogenetic analysis of HSV-2 G and 20 other HSV-2 strains. Since TR_L_ and IR_L_, TR_S_ and IR_S_ are two pairs of inverted repeat sequences, we used the genome sequences with TR_L_ (~9 K) and TR_S_ (~7 K) trimmed to perform phylogenetic analysis. These trimmed non-redundant sequences were aligned using ClustalW with standard parameters (v2.1) [41]. The alignment file was then converted to mega format with the MEGAX software package (v10.2.6) [42]. A maximum-likelihood tree was generated using MEGAX software, with the general time-reversible nucleotide substitution model with 5 gamma categories, 1000 bootstrap replicates, and complete deletion of alignment gaps, giving a total of 126,769 positions in the data set.

The following diverse HSV-2 strains were selected: MS (NCBI Accession: MK855052.1), HG52 (NCBI Accession: NC_001798.2), 333 (NCBI Accession: LS480640.1), SD90e (NCBI Accession: KF781518), strain 1192 (NCBI Accession: KP334095), strain COH 3818 (NCBI Accession: KP334096), CtSF (NCBI Accession: KP334097), CtSF-R (NCBI Accession: KP334093), GSC-56 (NCBI Accession: KP334094), 2007–38606 (NCBI Accession: KX574873.2), 2007–8222 (NCBI Accession: KX574883.2), JA2_JP (NCBI Accession: KR135323.1), J32715_UG (NCBI Accession: KR135315.1), 7444_1996_25809_US (NCBI Accession: KR135314.1), BethedaP5_US (NCBI Accession: KR135330.1), K39924_UG (NCBI Accession: KR135305.1), 44_419851_US (NCBI Accession: KR135309.1), SD66 (NCBI Accession: KR135320.1), and 9335_2007_14_US (NCBI Accession: KR135313.1). HSV-1 strain 17 (NCBI Accession: NC_001806.2) was used as an outgroup control.

## 3. Results

### 3.1. Genomic Sequencing, Assembly, and Annotation

HSV-2 G genomic DNA was sequenced with the PacBio platform and obtained a total of 1,166,537 subreads with mean read length of 7,644 nucleotides (nt). A total of seven contigs, with lengths of 136,152 bp, 38,474 bp, 33,551 bp, 38,035 bp, 34,083 bp, 31,680 bp, and 27,801 bp, respectively, were de novo assembled with PacBio HGAP4. The alignment of these 7 contigs against the genome sequence of HSV-2 reference strain HG52 (NCBI Accession: NC_001798.2 [27] by BLASR (v2.0.0) [35] with default settings showed that 2 contigs (Contig 1: 136,152 bp and Contig 2: 38,474 bp) jointly covered the full-length of the HG52 reference genome (Figure 1A). We determined the full-length of the ‘α’ sequence for strain G by comparing these contigs to the ‘α’ sequences retrieved from 3 other annotated complete HSV-2 genomic sequences: HG52, 333, and MS in NCBI GenBank. The detailed analysis of the sequence is described in a separate section below (Section 3.4). We merged contig 1 and 2 into a draft genome sequence and attached a full-length ‘α’ sequence to both ends. The draft sequence was then refined by the PacBio genome resequencing pipeline (SMRT^®^ Analysis v6.0) with default settings. It has been reported that HSV genomes have four isomers with two unique regions inverted corresponding to each other [25,29,43] and the assembly presented in this paper used the protype (P) isomer (Figure 1B) following convention. The termini of the genome were determined as described in a separate section below (Section 3.3). The finished length of the complete genome is 155,498 bp with 70.48% GC content. Out of 59,111 filtered CCS reads (read length > 1000 nt, mean read length of 9298 nt), a total of 13,496 CCS reads were mapped to the finished genome sequence with an average depth of 678X. The high coverage ensures high-quality consensus base calling (>99.999% accuracy) [32].

The genome of HSV-2 G has the same structure as HSV-1 and other HSV-2 genomes. Two unique regions (U_L_ and U_S_) and four repeat regions (TR_L_, IR_L_, IR_S,_ and TR_S_) flanking these two unique regions, as shown in Figure 1B. Using published complete genome sequences of HSV-2 strains HG52, 333, and MS as references, we annotated the finished HSV-2 G genomic sequence (NCBI Accession: OM370995). A total of 71 CDS were identified, including RL1 and RL2 as duplicates in the two inverted long repeats flanking U_L_ (TR_L_ and IR_L_) and RS1 as a duplicate in two inverted short repeats flanking U_S_ (IR_S_ and TR_S_). We also identified 17 HSV2 miRNA in HSV-2 G strain by aligning its genomic sequence to the HSV-2 miRNA stem_loop sequences obtained from the microRNA database miRBase with a keyword search of ‘hsv2′ (https://www.mirbase.org/ accessed on 19 August 2021). Among them, 4 (hsv2-mir-H11, hsv2-mir-H21, hsv2-mir-H22, and hsv2-mir-H23) are in U_L_, 9 (hsv2-mir-H2, hsv2-mir-H3, hsv2-mir-H4, hsv2-mir-H6, hsv2-mir-H7, hsv2-mir-H9, hsv2-mir-H19, hsv2-mir-H20, and hsv2-mir-H24) were identified within TR_L_ and IR_L_ as duplicate, and 4 (hsv2-mir-H5, hsv2-mir-H12, hsv2-mir-H13, and hsv2-mir-H25) were found in IR_S_ and TR_S_, with hsv2-mir-H12 and hsv2-mir-H13 found as duplicates in each repeat sequence, for a total of 4 copies. Furthermore, we located one *ori*L within U_L_ and two *ori*S within IR_S_ and TR_S_ (Figure 1B).

### 3.2. Confirmation of Four Isomers with ONT Long Reads

Investigators have used enzyme digestion to demonstrate that HSV genomes have four isomers with two unique regions inverted with respect to each other [24,25]. Recently, Lopez-Munoz et al. observed that the PacBio reads aligned to US1 or US12 can reach the IR_L_ region, and reads aligned to UL56 can span to the IR_S_ region. This observation is consistent with the four-isomers theory. However, the researchers could not find the reads spanning from U_L_ to U_S_ region [29]. In this study, we took advantage of the ONT to generate reads long enough to cover the full length of the internal repeat (IR) of the genome (IR_L_ and IR_S_ joined by ‘α’ sequence) flanked by U_L_ and U_S_. The region spanning the end of U_L_, the IR, and the beginning of U_S_ are unique for the four HSV-2 isomers of the HSV-2 genome. As such, UL56-IR-US1, UL56-IR-US12, UL1-IR-US1, and UL1-IR-US12 can be used to differentiate reads originating from P, I_S_, I_L,_ and I_SL_ isomer types (Figure 1B, boxed regions). We composed 4 sequences for these ULend-IR-USstart regions, spanning 1000 bp of U_L_ next to IR_L_ and 1000 bp of U_S_ next to IR_S,_ as references to which we aligned ONT sequencing reads. The reads mapping to UL56 and US1 with length >500 bp and identity >80% were counted as the P-type genome, and the reads mapping to UL56 and US12, UL1 and US1, UL1, and US12 with the same criteria were considered to support I_S_, I_L,_ and I_SL_-type of genomes, respectively. As a result, 76, 102, 82, and 74 reads were identified to support P, I_S_, I_L,_ and I_SL_ isomers, respectively. To our knowledge, this is the first time that the four isomers of the HSV-2 genome were observed directly at the nucleotide level, and this result is consistent with the previous assessment of four equimolar isomers of HSV genomes.

### 3.3. Determination of Genome Termini of HSV-2 G

Due to the repetitive sequences and high GC content, it is difficult to precisely define the termini of the HSV genome. As a result, the reported termini sequences for HSV-2 have been either missing or have discrepancies between different reports. Long reads can span the repetitive regions in the HSV genome and precisely determine the termini of the genome. The unique genes closest to the termini were used to align and select the reads that contain the termini at the end. To determine the 5′ terminus of the genome sequence, the CCS reads aligning to the first 500 bp of the UL1 gene were selected and aligned to the P type HSV-2 G genome sequence (Figure 1B). The majority of 5′ terminus reads start aligning to our final genome sequence at the second position (Figure 2A,B). We also observed that some reads belonging to this terminus contained two and more copies of the ‘α’ sequence. There are also reads extending to the IR_S_/TR_S_ sequence which could have been generated from the U_L_-U_S_ joint or link termini within a circular genome. Similarly, to define the 3′ terminus of the genome sequence, the CCS reads aligned to the first 500 bp of the US12 gene were selected and aligned to the P type HSV-2 G genome sequence (Figure 1B). The alignment of the majority of 3′ terminus reads ended at the second to last position in our final genome sequence (Figure 2C,D). We observed that some reads with more than one copy of the ‘α’ sequence either extended the TR_L_/IR_L_ region or the outside ‘α’ sequence is not whole ‘α’ sequence, indicating that they are not terminus reads. The reason that we added an extra nucleotide beyond the read alignment for 5′ and 3′ termini will be explained in the next section due to the nature of the ‘α’ sequence in HSV-2 genome and the PacBio sequencing procedure.

### 3.4. Analysis of ‘α’ Sequences of HSV2 Strain G

To analyze the ‘α’ sequences of HSV-2 G, we retrieved and aligned the ‘α’ sequences between IR_L_ and IR_S_ from 10 HSV-2 strains in NCBI GenBank and strain G from this study. Alignment of these 11 ‘α’ sequences showed that all of them were bracketed by a conserved 15 bp DR1 sequence in the same direction (Figure 3). The shortest ones (267 bp) are ‘α’ sequences in HSV2 strains 2007-38606 and SD90e, and the longest (425 bp) is from the HSV-2 MS strain. Conserved package motifs (Pac1 T, Pac2 Consensus, and Pac2 T) have been identified in the HSV-1 DR1 sequence, and they participate in the virus packaging process of HSV-1 [40]. We have identified the sequence with high homology to these motifs in all ‘α’ sequences of the 11 HSV-2 strains, and they are located in a similar context as that of HSV-1.

The ‘α’ sequence between IR_L_ and IR_S_ is in the inverted direction of the ‘α’ sequence at the two termini (Figure 1B). Three copies of the assembled ‘α’ sequences in this HSV-2 G genome sequence were aligned using the reverse complement of the ‘α’ sequence between IR_L_ and IR_S_ (Figure 4A). We observed that 5′ terminus reads start aligning at the second position of the reversed ‘α’ sequence within the DR1 region and 3′ terminus reads end alignment at the 16th position if counting from the end of a reversed ‘α’ sequence (lacks 15 nucleotides) (Figure 2). This phenomenon arises from HSV-2 genome replication. It is reported that the HSV-1 genome has one 3′ overhang nucleotide at both termini and lacks 18 nucleotides at the 3′ terminus. As a result, the two copies of ‘α’ sequence will form a shared DR1 sequence when two termini of the linear genome are ligated to form the circular form of the genome. After the genome is replicated with the rolling-circle model and produces concatenated multiple genomes, endonuclease G will cut the concatenated multiple genomes within DR1, producing two 3′ overhang nucleotide termini and the 3′ genome terminus misses 18 nucleotides [44]. We hypothesize that HSV-2 G has a similar genome replication process (Figure 4B). The 3′ overhanging A at the 5′ terminus and T at the 3′ terminus had been removed before sequencing adaptors were linked with the DNA fragment during sequencing library preparation; therefore, the reads start aligning from the second position of both termini. The DR1 is 15 nucleotides of length in HSV-2; thus, the 3′ terminus lacks 14 nucleotides. The finished sequence was based on this alignment information and we included the 3′ overhanging base at each terminus. In HSV-1, the enzyme involved in this process is endonuclease G; the enzyme involved in the HSV-2 genome replication process needs to be further investigated, given the difference of DR1 sequences between HSV-1 and HSV-2.

### 3.5. Determination of ‘α’ Sequence Copy Number in HSV2 Strain G Genome

It was reported in the literature that only one copy of ‘α’ sequence was observed in the 3′ TR_S_ terminus, but more than one consecutive copy could exist in the 5′ TR_L_ terminus and the joint location of internal R_L_ and R_S_ (Figure 1B). The ‘α’ sequences of HSV-2 are bracketed by the DR1 sequence and the two neighboring ‘α’ sequences have a shared DR1 if two or more tandem repeats of ‘α’ sequence are at the 5′ terminus and between the IR_L_ and IR_S_.

To systematically determine the copy number of consecutive ‘α’ sequences, we aligned all CCS reads to the ‘α’ sequences of HSV-2 G and found 927 CCS reads containing ‘α’ sequences with mapped length greater than 200 bp. To further determine which reads are in TR_L_, TR_S,_ or IR regions, we aligned these 927 reads to the RL1 and RS1 genes. Reads containing both RL1 and RS1 with mapping length >200 bps are considered as IR reads because they join the internal repeat region R_L_ and R_S_. Reads containing only RL1 (>200 bps) but not RS1 and starting or ending with a nearly full-length ‘α’ sequence (>360 bps) are considered as TR_L_ reads, and similarly, reads containing only RS1 (>200 bps) but not RL1 and starting or ending with a TRs ‘α’ sequence (349 bps) are considered as TRs reads. In total, we identified 308 IR reads, 223 TR_L_ reads, and 145 TRs reads meeting these criteria. Among the 223 TR_L_ reads, 157 of them (~70%) contain a single copy, 53 reads (24%) have 2 copies, 6 reads (3%) have 3 copies, and 7 reads (3%) have 4 copies of ‘α’ sequence. Similarly, among 308 IR reads, 153 of them (50%) contain a single copy, 123 reads (40%) have 2 copies, 12 reads (4%) have 3 copies, and 20 reads (6%) have 4 copies of ‘α’ sequence. In contrast, all 145 TR_S_ reads have a single copy of ‘α’ sequence (Table 1).

### 3.6. Comparison of Genome Sequences between HSV-2 Strains and Phylogenetic Analysis of HSV2

We have compared three published complete HSV-2 genome sequences, including strain HG52, 333, and MS. We have identified various mutations within these highly homologous genome sequences (Table 2 summarizes the results, and mutation details are listed in Appendix A. Compared with strain G, there are substitution mutations in 65, 57, and 57 genes in HSV-2 strain MS (Appendix A), HG52 (Appendix A), and 333 (Appendix A), respectively. They also have insertion/deletion (indel) mutations in these genomes (Appendix A). There is a 54 bp deletion within US11 gene in strain G genome. This deletion leads to the loss of the stop codon and causes a frameshift starting at AA position 144 (second to last AA), adding eleven additional amino acids to the end (Figure 5).

To further evaluate the relationship of HSV-2 G with other HSV-2 strains, we have performed phylogenetic analysis with 20 other HSV-2 strains and HSV-1 strain 17 as an outgroup control (Figure 6A,B, Appendix A). Notably, some nodes have a low bootstrapping value. However, the relationships between strains are consistent with the results of phylogenetic analyses in the published literature [29,45,46,47,48], suggesting the phylogenetic tree reported here is reasonable. HSV-2 G is close to HSV-2 strain 1192, which was isolated from a genital lesion in Wisconsin, USA, and was sequenced by Kolb et al. [46]. The next closest strains are HSV2 strain 9335-2007-14, GSC-56, and HG52 [27]. All of them originated from the US or UK, which has close social and economic connections with the US. The furthest strains from strain G were 2007–38606, which originated from Peru; J32715, which originated from Uganda; and 2007-8222, which originated from Zambia. Overall, the HSV-2 strains which originated from the US and Europe are closer to strain G than HSV-2 strains that originated from other regions (Table 3 and Appendix A).

## 4. Discussion

The challenge of the treatment for HSV-2 infection is recurrent reactivation of the HSV-2 virus. Many HSV-2 strains have been identified and have different biological properties. Complete genome sequences of these strains will significantly advance clinical research of HSV-2. However, high GC content and highly repetitive regions in the HSV-2 genome make it difficult to obtain complete genome sequences. In this study, we have determined the complete genomic sequence of HSV-2 G with PacBio sequencing data. It is the fifth complete genome sequence of an HSV-2 strain. The finished length of the complete genome is 155,498 bases with 70.48% GC content. A total of 71 coding sequences and 17 miRNAs have been identified. Using ONT sequencing, we confirmed, for the first time, the existence of four genome isomers at the nucleotide level. The ‘α’ sequence and genome termini have been investigated in detail. The HSV-2 packaging signal motifs and cleavage site within ‘α’ sequence have been identified.

Taking advantage of the length of PacBio sequencing reads, we can visualize genome termini of HSV-2 G by selecting the reads aligned to UL1 and US12, aligning to the assembled genome, and visualizing in IGV (Figure 2). Similar to reported HSV-1 termini, we observed single T 3′ overhangs at the TR_L_ end and single A 3′ overhangs at the TR_S_ end of the viral genome, respectively. At the TR_S_ end, we observed only one copy of ‘α’ sequence with one DR1 missing at the very end except a 3′ protruding A (Figure 4B). It is suggested that two 3′ single nucleotide overhang termini were generated by endonuclease G in HSV-1 [44]. However, the DR1 sequence in the HSV-1 genome is different from that of HSV-2, indicating that a different endonuclease may be involved in the process of concatenated multiple genomes to produce single linear HSV-2 genome units and needs further investigation. The involved endonuclease can be a drug target for HSV-2 treatment and control. After aligning and analyzing ‘α’ sequences of 11 HSV-2 strains, including strain G in this study, we identified the packaging motifs and observed that they have high homology to that of HSV-1, including their flanking regions (Figure 3), indicating that HSV-2 has similar cleavage and packaging mechanisms to HSV-1. Studying the protein interacted with these motifs could lead to the discovery of the novel proteins/enzyme system involved in HSV genome processing and packaging. Since we could distinguish the PacBio sequencing reads of the TR_L_ end, IR_L_-IR_S_ joint, and TR_S_ end, we have been able to precisely determine the copy number of ‘α’ sequence at the TR_L_ end, IR_L_-IR_S_ joint, and TR_S_ end (Table 1). Like HSV-1 [43], HSV-2 can have one or more copies of ‘α’ sequence at the TR_L_ end and IR_L_-IR_S_ joint, and one copy of partial ‘α’ sequence at the TR_S_ end. The mechanism of controlling the copy number of ‘α’ sequence is unknown and needs further investigation.

Phylogenetic analysis of the complete HSV genome requires a lot of computational resources. To reduce the computational burden, we have trimmed TR_L_ and IR_S_ before analysis because they are the same sequence of IR_L_ and IR_S_. We have used diverse HSV-2 strains from several studies [29,45,46,47,48]. One benefit of this design is that it will provide an evaluation reference of our phylogenetic tree (Figure 6). The strains used have a relationship consistent with other studies; we concluded that our tree is reasonable even though several nodes in our phylogenetic tree have low bootstrapping values. Among the strains analyzed, we have identified that the closest HSV-2 strain to strain G is strain 1192, which originated from Wisconsin in the US, and the furthest strain to strain G is 2007-38606, which originated in Peru. Overall, the HSV-2 strains that originated from the US and Europe are closer to strain G than HSV-2 strains that originated from other regions (Table 3 and Appendix A).

The CDS of HSV-2 strain G have been compared with those of strain MS, HG52, and 333. The majority of CDS have substitution and indel mutations (Table 2 and Appendix A). These differences could contribute to the biological differences of these strains. At the same time, the differences within non-coding regions could also contribute to the biological differences.

In summary, we have assembled the complete genome sequence of HSV-2 G using the PacBio sequencing platform. The sequence has been annotated, and important genome structures have been determined. The relationship with diverse HSV-2 strains has been determined by phylogenetic analysis. The ‘α’ sequence of HSV-2 has been analyzed and the virus DNA cleavage site and packaging signal motifs have been identified. Identifying the protein factor interacting with these regions will reveal the virus DNA processing and packaging mechanism. In conclusion, the complete genome sequence of HSV-2 G can help future clinical research and treatment of HSV-2 infection.

## Figures and Tables

**Figure 1 viruses-14-00536-f001:**
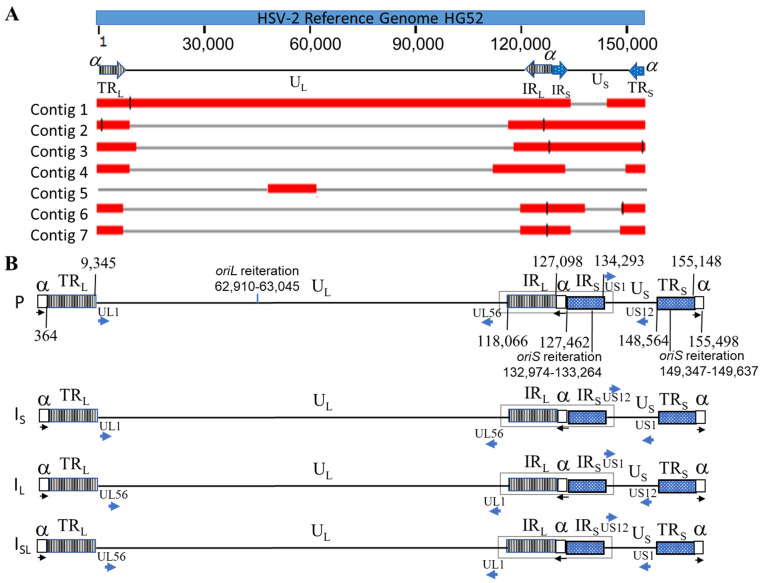
HSV-2 genome and alignments of seven contigs. (**A**) The alignments of the seven contigs against the reference genome HSV-2 strain HG52. Nucleotide lengths of contig 1, 2, 3, 4, 5, 6, and 7 are 36,152 bp, 38,474 bp, 33,551 bp, 38,035 bp, 34,083 bp, 31,680 bp, and 27,801 bp, respectively. (**B**) Four isomers of HSV-2 strain G genome: black arrow: ‘α’ sequence orientation; blue arrow: gene orientation. The sequences of boxed regions were used as refence for identification sequencing reads generated from four isomers.

**Figure 2 viruses-14-00536-f002:**
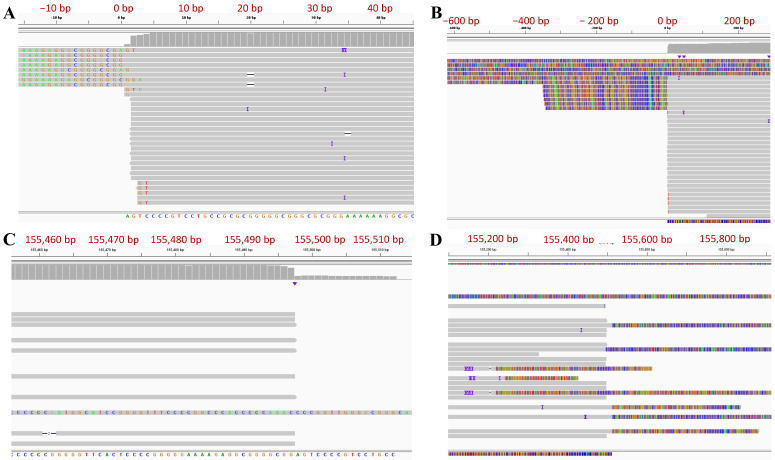
Genome Termini of HSV-2 strain G (P type) shown in IGV. (**A**) Zoomed in and (**B**) Zoomed out of 5’ terminus of the Genome. (**C**) Zoomed in and (**D**) Zoomed out of 3’ terminus of the Genome. The nucleotide positions of HSV-2 strain G genome sequence were labeled along the alignment and they have been mirrored by the numbers in red for better readability.

**Figure 3 viruses-14-00536-f003:**
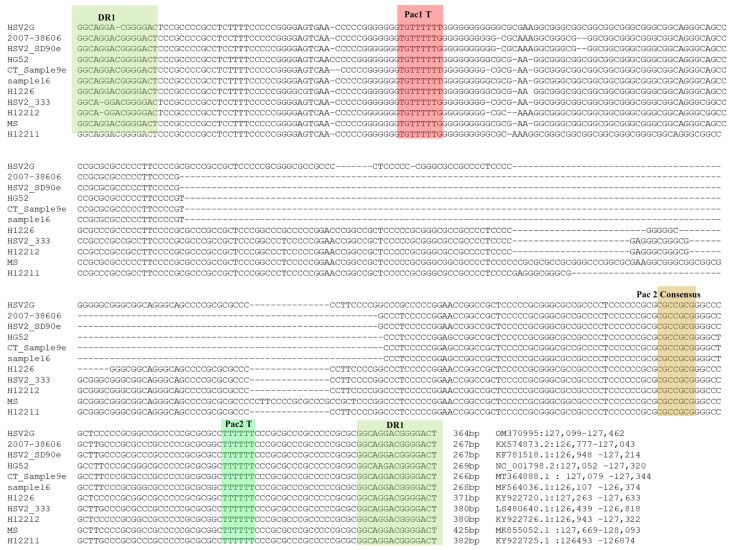
The alignment of ‘α’ sequences between IRL and IRS from HSV-2 G and 10 other strains. The packaging signal motifs homologous to HSV-1 [1] were highlighted.

**Figure 4 viruses-14-00536-f004:**
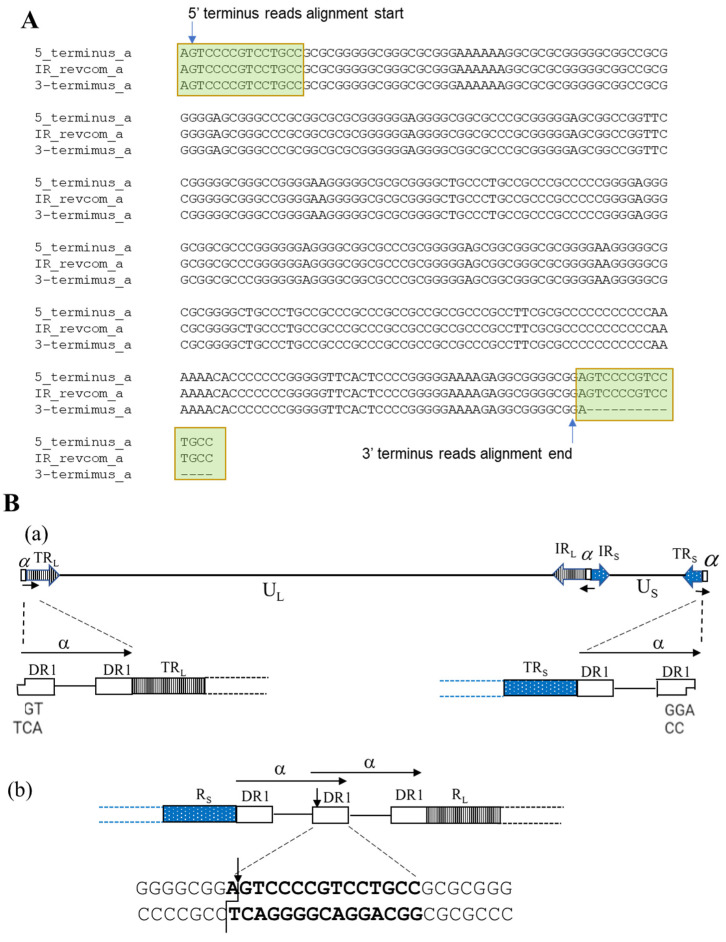
Analysis of HSV-2 genome termini: (**A**) Alignment of ‘α’ sequence of both termini and ‘α‘ sequence between IRL and IRS in reverse orientation. The terminus reads’ alignment positions are labeled; (**B**) Proposed HSV-2 circularization from two 3’-overhang termini (**a**) and formation of the termini after genome DNA replication by cleavage of the concatenated multiple genomes (**b**).

**Figure 5 viruses-14-00536-f005:**
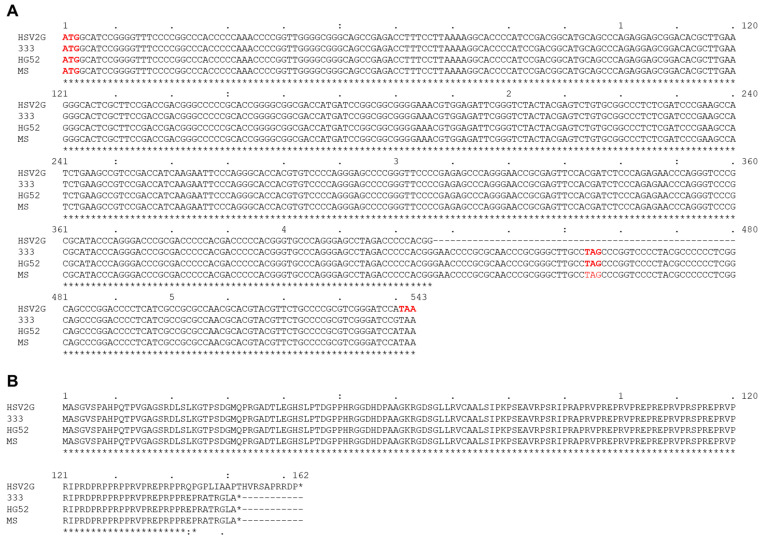
The alignment of US11 coding sequence (**A**) and protein sequence (**B**) of HSV-2 strain G with strain 333, HG52, and MS. The deletion of 54 nt around the stop codon in strain G leads to a frameshift of the last nine amino acids adding additional eleven amino acids to the end. The conserved positions are denoted with *.

**Figure 6 viruses-14-00536-f006:**
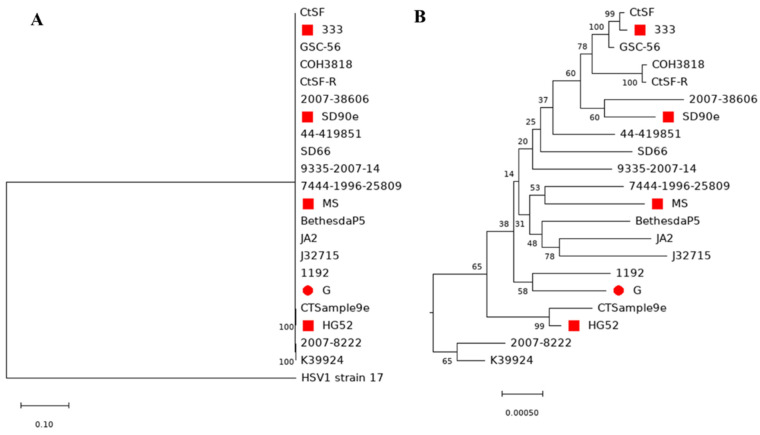
Phylogenetic tree of 20 HSV-2 genome sequences including the new genome sequence of strain G from this study. (**A**) Maximum-likelihood tree generated with MEGAX using 1000 bootstrap replicates. HSV-1 strain 17 was used as an outgroup control. (**B**) Zoomed in visualization of the HSV-2 cluster. Bootstrapping values in percentage were indicated for each node. The red circle labels strain G in the current study, and red squares mark the published complete HSV2 genome sequence from NCBI.

**Table 1 viruses-14-00536-t001:** Read Count of ‘α‘ Sequence Copy Number at Three Different Location.

αSeq_Read_Location	Total Reads	1 αSeq	2 αSeq	3 αSeq	4 αSeq
aSeq_m_-TRL	207	171	31	3	2
IRL-aSeq_n_-IRs	196	121	62	11	2
aSeq(349)-TRs	145	145	0	0	0

**Table 2 viruses-14-00536-t002:** Comparison of CDS of Three Complete HSV-2 Genome with HSV-2 Strain G.

Strain	NCBI Accession	Genome Length	Substitution Only	In/Del Only	Substitution and In/Del	Frameshift
MS	MK855052.1	155,975	52	1	13	1
HG2	NC_001798.2	154,675	44	1	13	1
333	LS480640.1	155,503	43	0	14	1

Note: In: Insertion; Del: Deletion.

**Table 3 viruses-14-00536-t003:** The HSV-2 Strains Used in Phylogenetic Analysis and Their Distances to Strain G.

Virus	Strains	Distance to G	Origin Location	NCBI Accession	Sequence Ref
HSV-2	1192	0.001889	Wisconsin, United States	KP334095.1	[46]
HSV-2	9335-2007-14	0.002367	Seattle, WA, USA	KR135313.1	[47]
HSV-2	GSC-56	0.002409	United States	KP334094.1	[46]
HSV-2	HG52	0.002414	Scotland, United Kingdom	NC_001798.2	[27]
HSV-2	44-419851	0.002414	MD, USA	KR135309.1	[47]
HSV-2	7444_1996_25809	0.002517	Seattle, WA, USA	KR135314.1	[47]
HSV-2	CtSF	0.002530	United States	KP334097.1	[46]
HSV-2	333	0.002567	Texas, United States	LS480640.1	[28]
HSV-2	BethesdaP5	0.002605	MD, USA	KR135330.1	[47]
HSV-2	SD66	0.002619	Carletonville, South Africa	KR135320.1	[47]
HSV-2	MS	0.002784	Reykjavik, Iceland	MK855052.1	[29]
HSV-2	K39924	0.002797	Rakai, Uganda	KR135305.1	[47]
HSV-2	CT_Sample9e	0.002799	Seattle, WA, USA	MT364888.1	[49]
HSV-2	CtSF-R	0.002800	United States	KP334093.1	[46]
HSV-2	COH3818	0.002809	United States	KP334096.1	[46]
HSV-2	JA2	0.002870	Japan	KR135323.1	[47]
HSV-2	SD90e	0.002915	Carletonville, South Africa	KF781518.1	[30]
HSV-2	2007–8222	0.003046	Zambia	KX574883.2	[48]
HSV-2	J32715	0.003062	Rakai, Uganda	KR135315.1	[47]
HSV-2	2007–38606	0.003264	Peru	KX574873.2	[48]
HSV-1	strain_17	0.573289	Scotland, United Kingdom	NC_001806.2	[50]

## Data Availability

The complete HSV-2 strain G genome sequence has been deposited in GenBank under the accession number OM370995; PacBio Sequencing CCS reads and ONT Sequencing data were deposited in the SRA with the accession number SRR17709760 and SRR17709759, respectively.

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
