# Peer review of "Complete Genome Sequence of Herpes Simplex Virus 2 Strain G"

_viruses, 2022, doi:10.3390/v14030536_

Round 1

Reviewer 1 Report

Minor points

A sound study demonstrating the power of long read sequencing of higly repetetive regions of high GC.

My only point is that figures need attention in formatting style and size of text.

Line 148 ‘a’

Figure 1. Missing grey line on Contig 5

Figure 2D, uncertain what the roman numerals refer to

All figures, some text in in very small letters, hardly readable

L283/296  ”start aligning”

Figure 5, include amino acids

Author Response

A sound study demonstrating the power of long read sequencing of higly repetetive regions of high GC.

My only point is that figures need attention in formatting style and size of text.

Authors’ response: Thank you for your nice comments. The following is our point-to-point response and correction.

Line 148 ‘a’

Authors’ response: Thanks. Reformatted to “Symbol”.

Figure 1. Missing grey line on Contig 5

Authors’ response: Thanks. The grey line was added.

Figure 2D, uncertain what the roman numerals refer to

Authors’ response: Thanks. Added a legend to explain the number. These numbers are the position of the reference sequence.

All figures, some text in in very small letters, hardly readable

Authors’ response: Thanks. All figures have been modified as follows to make the text larger:

Figure 1: Increased the size to make all labels legible.

Figure 2: The figure size was increased. The numbers from the screenshot of IGV are too small to read and are mirrored by the new label with a different color (Red).

Figure 3: We decreased the nucleotide number on each line; the font size was then increased.     

Figure 4: We rearranged the panel and increased each panel size to increase the font size.

Figure 5: We shortened the ID of the sequences and increased the font size.

Figure 6: We increased the figure size and the font size to be large enough to read afterwards.

L283/296  ”start aligning”

Authors’ response: Thank you for pointing this out. They are now corrected.

Figure 5, include amino acids

Authors’ response: Thanks. We have added Panel B with the protein sequence alignment. 

A sound study demonstrating the power of long read sequencing of higly repetetive regions of high GC.

My only point is that figures need attention in formatting style and size of text.

Authors’ response: Thank you for your nice comments. The following is our point-to-point response and correction.

Line 148 ‘a’

Authors’ response: Thanks. Reformatted to “Symbol”.

Figure 1. Missing grey line on Contig 5

Authors’ response: Thanks. The grey line was added.

Figure 2D, uncertain what the roman numerals refer to

Authors’ response: Thanks. Added a legend to explain the number. These numbers are the position of the reference sequence.

All figures, some text in in very small letters, hardly readable

Authors’ response: Thanks. All figures have been modified as follows to make the text larger:

Figure 1: Increased the size to make all labels legible.

Figure 2: The figure size was increased. The numbers from the screenshot of IGV are too small to read and are mirrored by the new label with a different color (Red).

Figure 3: We decreased the nucleotide number on each line; the font size was then increased.     

Figure 4: We rearranged the panel and increased each panel size to increase the font size.

Figure 5: We shortened the ID of the sequences and increased the font size.

Figure 6: We increased the figure size and the font size to be large enough to read afterwards.

L283/296  ”start aligning”

Authors’ response: Thank you for pointing this out. They are now corrected.

Figure 5, include amino acids

Authors’ response: Thanks. We have added Panel B with the protein sequence alignment. 

Reviewer 2 Report

The paper written by Chang et al. reports the sequencing and bioinformatics analysis of the complete genome of HSV2. An herpersvirus that is pathogenic for humans and with a genital tropism. This is the 5th strain sequenced and available in public databases. The comparative analysis of the strains shows extremely close genomes as often in genomic analysis of the Herperviridae family.

This HSV2 G strain has an important interest in medical research.

Sequencing using the combination of the Pacbio and ONT technique is very interesting and increasingly informative. In this paper, it allows the detection of isoforms for the first time. Nevertheless the description of these isoforms can be more precise? synonyms can be used as viral subpopulations ? When can we distinguish what is an isomer from what is a viral subpopulation?

Some points of this paper need to be improved.

In the abstract a non-specialist cannot understand the Alpha sequences

This comment is also valid in the introduction where no elements are provided by the authors.

In the material and method section 2.1

Why remove reads smaller than 3kb? More rational or a reference should be added.

What are the parameters used during the 65 hours of sequencing? MUX scan every how many? more data can be provided by the authors.

In part 2.3 the authors explain how they finished the HSV2 sequences.

Was there a big difference between the results of the assembly and what was the contribution of the sequencing? And related to the section 3.3 :

Line 171 page 4 you have 2 comas ",,"

Line 191 " it has been reported " where is the reference?

The table 2 could be rather a venn diagram to see a real genomic comparison.

Concerning low bootstrapping values the authors can use other phylogeny methods in addition especially for the calculation of bootstrapping? Maybe use fastree? Use a GTR for close phylogenomic sequences explain the bootstrap maybe remove branches with score under 50.

Author Response

The paper written by Chang et al. reports the sequencing and bioinformatics analysis of the complete genome of HSV2. An herpersvirus that is pathogenic for humans and with a genital tropism. This is the 5th strain sequenced and available in public databases. The comparative analysis of the strains shows extremely close genomes as often in genomic analysis of the Herperviridae family.

This HSV2 G strain has an important interest in medical research.

 Authors’ response: Thank you for your nice comments. The following is our point-to-point response and correction.

Sequencing using the combination of the Pacbio and ONT technique is very interesting and increasingly informative. In this paper, it allows the detection of isoforms for the first time. Nevertheless the description of these isoforms can be more precise? synonyms can be used as viral subpopulations ? When can we distinguish what is an isomer from what is a viral subpopulation?

 Authors’ response: Thank you for your nice comments. Isomer is the term used to describe the four different arrangements of UL and US in the genome of the HSV samples. They exist in equal amounts and have the same sets of genetic material due to the two inverted repeats flanking UL and US. On the other hand, viral subpopulations have different genotypes due to mutations or recombination.  

Some points of this paper need to be improved.

In the abstract a non-specialist cannot understand the Alpha sequences

Authors’ response: Thanks. We added a brief introduction of the Alpha sequence in the abstract.

This comment is also valid in the introduction where no elements are provided by the authors.

 Authors’ response: Thanks. We added more information for the Alpha sequence in the introduction.

In the material and method section 2.1

Why remove reads smaller than 3kb? More rational or a reference should be added.

Authors’ response: Thanks. Removal of shorter insert fragments that would be preferentially sequenced if present helps with achieving a larger subread mean length which can help with assembly as well as improving total sequencing yield. We added a sentence to provide the explanation.

What are the parameters used during the 65 hours of sequencing? MUX scan every how many? more data can be provided by the authors.

 Authors’ response: Thanks. The Nanopore sequencing details provided in the manuscript are on par with information commonly found in the literature. Default settings were used for the sequencing run including 90 min mux scans. We added this information into manuscript as you suggested.

In part 2.3 the authors explain how they finished the HSV2 sequences.

Authors’ response: Thank you for your suggestion. We have added the suggested information in 2.3 and 3.4.

Was there a big difference between the results of the assembly and what was the contribution of the sequencing? And related to the section 3.3 :

Authors’ response: Thanks. Yes. there are about 2,000 bp mismatches and more than 1,000 bp gaps distributed along the genome by comparing the sequence obtained by overlapping contig 1 and contig 2 (Figure 1) with finished HSV-2 strain G sequence. Since it is standard workflow, we won’t add any information here.

Line 171 page 4 you have 2 comas ",,"

 Authors’ response: Thank you for catching this. It has been corrected.

Line 191 " it has been reported " where is the reference?

 Authors’ response: Thank you for your suggestion. The references have been added.

The table 2 could be rather a venn diagram to see a real genomic comparison.

 Authors’ response: Thank you for your suggestion. We agree that a Venn diagram would be more visual. However, the Venn diagram which convey the complex information in Table 2 is very complex. Therefore, we don’t replace this table with venn diagram.

Concerning low bootstrapping values the authors can use other phylogeny methods in addition especially for the calculation of bootstrapping? Maybe use fastree? Use a GTR for close phylogenomic sequences explain the bootstrap maybe remove branches with score under 50.

Authors’ response: Thank you for your comments and suggestion. The low bootstrapping values couldn’t be improved by using other methods. We couldn’t improve the low bootstrapping value after removing up to four branches either. However, as we discussed in our manuscript, the strains used in our analysis study have relationship consistent with other published studies. Therefore, the phylogenetic tree presented in the manuscript is reasonable.